# Food Preference Assessed by the Newly Developed Nutrition-Based Japan Food Preference Questionnaire and Its Association with Dietary Intake in Abdominal-Obese Subjects

**DOI:** 10.3390/nu16234252

**Published:** 2024-12-09

**Authors:** Naoko Nagai, Yuya Fujishima, Chie Tokuzawa, Satoko Takayanagi, Mikiko Yamamoto, Tomoyuki Hara, Yu Kimura, Hirofumi Nagao, Yoshinari Obata, Shiro Fukuda, Megu Y. Baden, Junji Kozawa, Norikazu Maeda, Hitoshi Nishizawa, Iichiro Shimomura

**Affiliations:** 1Department of Metabolic Medicine, Graduate School of Medicine, Osaka University, 2-2, Yamada-oka, Suita 565-0871, Osaka, Japan; nagaink@hosp.med.osaka-u.ac.jp (N.N.); t-hara@endmet.med.osaka-u.ac.jp (T.H.); kimura-u@endmet.med.osaka-u.ac.jp (Y.K.); nagao@endmet.med.osaka-u.ac.jp (H.N.); obata-yoshinari@endmet.med.osaka-u.ac.jp (Y.O.); s.fukuda@endmet.med.osaka-u.ac.jp (S.F.); mbaden@endmet.med.osaka-u.ac.jp (M.Y.B.); kjunji@endmet.med.osaka-u.ac.jp (J.K.); norikazu_maeda@med.kindai.ac.jp (N.M.); ichi@endmet.med.osaka-u.ac.jp (I.S.); 2Division of Nutritional Management, Osaka University Hospital, 2-15, Yamada-oka, Suita 565-0871, Osaka, Japan; chienish@hosp.med.osaka-u.ac.jp (C.T.); tsatoko@hosp.med.osaka-u.ac.jp (S.T.); mk5hl04@hosp.med.osaka-u.ac.jp (M.Y.); 3Department of Lifestyle Medicine, Graduate School of Medicine, Osaka University, 2-2, Yamada-oka, Suita 565-0871, Osaka, Japan; 4Department of Diabetes Care Medicine, Graduate School of Medicine Osaka University, 2-2, Yamada-oka, Suita 565-0871, Osaka, Japan; 5Department of Endocrinology, Metabolism and Diabetes, Faculty of Medicine, Kindai University School of Medicine, 377-2, Ohno-higashi, Osaka-Sayama 589-8511, Osaka, Japan; 6Department of Metabolism and Atherosclerosis, Graduate School of Medicine Osaka University, 2-2, Yamada-oka, Suita 565-0871, Osaka, Japan

**Keywords:** food preference, questionnaire, dietary intake, obesity, metabolic syndrome

## Abstract

Background/Objectives: Understanding food preferences is important for weight management. However, methods for assessing food preferences are not well established, especially in Japan. This study aimed to examine detailed food preferences and their associations with actual food intake in non-obese and abdominal-obese subjects using a newly developed questionnaire tailored for the Japanese population. Methods: We developed the Japan Food Preference Questionnaire (JFPQ) to evaluate food preferences across four nutrient groups based on nutritional evidence: carbohydrate, fat, protein, and dietary fiber. A total of 38 non-obese and 30 abdominal-obese participants completed both the JFPQ and the Food Frequency Questionnaire (FFQ). Food preferences for each nutrient were compared between the two groups, and correlations between food preferences (assessed by the JFPQ) and food intake (assessed by the FFQ) were analyzed. Results: Compared with the non-obese group, the abdominal-obese group showed significantly greater preferences for carbohydrates, fat, and protein, with no significant difference in dietary fiber after adjusting for age and sex. Furthermore, in the abdominal-obese group, positive correlations were found between actual intake and preference for high-fat and high-carbohydrate foods. Conclusions: Our findings from this pilot study demonstrated that abdominal-obese individuals had greater preferences for fat and carbohydrates, which were linked to actual fat and carbohydrate intake and possibly contributed to the development of obesity.

## 1. Introduction

The prevalence of obesity and obesity-related disorders is rapidly becoming a global health concern, particularly in East Asian countries, including Japan [1,2]. While weight loss through energy intake restriction is an effective treatment for obesity [3,4,5], managing the eating behavior of obese individuals is often challenging. Previous studies have demonstrated that Japanese individuals with obesity or visceral fat accumulation exhibit abnormalities in several eating behaviors, including food preference and content [6,7,8].

Among the multiple factors that influence food intake, food preferences are of primary importance in determining the choice, quantity, and content of foods consumed, which can contribute to body weight status as well as the development of chronic metabolic diseases, such as diabetes mellitus [9]. It has been reported that obese individuals exhibit a high preference for fat [10,11,12,13,14,15,16] and possibly sweet foods [10,12,14,15,16,17]. However, owing to the lack of appropriate tools based on nutritional evidence, the differences in food preferences for each nutrient between obese and non-obese individuals are not fully understood, especially in Japanese individuals. Moreover, clarifying the relationship between food preferences and actual food intake may help explain the underlying causes of obesity and visceral fat accumulation.

When exploring the relationship between an individual’s food preferences and the pathogenesis of obesity, it is crucial to recognize that food preferences are strongly influenced by the food culture, traditions, and eating habits of each country. Therefore, survey methodologies that account for these cultural factors on a country-by-country basis are needed. In particular, to appropriately assess the food preferences of the Japanese population, it is necessary to employ a nutrition-based questionnaire tailored to their specific eating habits and dietary composition. However, such a method has not yet been well established.

In this study, we developed a new food preference questionnaire based on detailed nutritional characteristics consisting of food items ordinally eaten by Japanese people. Using our questionnaire, the present study aimed to evaluate the differences in food preferences between non-obese and abdominal-obese individuals and to examine the relationships between these preferences and actual dietary intake.

## 2. Materials and Methods

### 2.1. Requirements

The present study enrolled non-obese and abdominal-obese individuals. The non-obese group was recruited from physicians and nutritionists between the ages of 20 and 74 years who were willing to participate in the study. The criteria for being classified as non-obese included a body mass index (BMI) less than 25 kg/m^2^; waist circumference less than 85 cm for men and 90 cm for women; and no medication for hypertension, dyslipidemia, or diabetes mellitus. The abdominal-obese group was recruited from a previously described feasibility pilot study of a lifestyle modification program [18], which included Japanese company employees aged 20 to 74 years who met the criteria of BMI ≥25 kg/m^2^ or waist circumference ≥85 cm for men and ≥90 cm for women, corresponding to a visceral fat area of 100 cm^2^ [19,20], as indicated at the annual health check-up. Those on special diets, such as protein restriction or elimination of allergic foods, those who were pregnant or breastfeeding, and those with severe hypertension (systolic blood pressure ≥160 mmHg or diastolic blood pressure ≥110 mmHg), severe hypertriglyceridemia (≥1000 mg/dL), and exercise restrictions (e.g., those with heart failure and/or unstable angina) were excluded from this study.

The current study was conducted in accordance with the Declaration of Helsinki and was approved by the Institutional Ethics Review Board of Osaka University Hospital (approval numbers 21387 and 21335). Written informed consent was obtained from all participants included in this study.

### 2.2. Development of the Japan Food Preference Questionnaire

To quantitatively assess food preference, the Japan Food Preference Questionnaire (JFPQ) was newly developed according to the following procedures: (1) Foods that are frequently discussed in nutrition counseling, such as weight loss guidance and blood glucose management for diabetic patients, were selected as candidates for the JFPQ. (2) The nutritional characteristics of the candidate foods were then investigated in detail by referring to the energy and nutrient values listed for each food in the “Standard Tables of Food Composition in Japan 2020 (8th Revised Edition)” [21] and “Tables of Carbohydrate Composition in Japan 2020 (8th Revised Edition)—Available Carbohydrates, Polyols, Dietary Fiber, Organic Acids” [22]. (3) Candidate foods were classified into carbohydrate, fat, and protein groups by defining a carbohydrate-to-total energy ratio of ≥70% as the carbohydrate group, a fat-to-total energy ratio of ≥40% as the fat group, and a protein-to-total energy ratio of ≥30% as the protein group. For the carbohydrate and fat groups, each food was further divided into two subgroups: a sweet subgroup with more than 10 g of total sugars, including monosaccharides (glucose, fructose, and galactose) and disaccharides (sucrose, maltose, lactose, and trehalose), per 100 g of edible portion, and a non-sweet subgroup with less than 10 g of total sugars. In addition, foods with a high dietary fiber content (≥1 g per 100 g of edible portion) and low energy (<100 kcal per 100 g of edible portion) were selected for the dietary fiber group. Fruits were included in both the carbohydrate and dietary fiber groups because they met the criteria for both groups. (4) In addition to the foods that were used to categorize each nutritional group, various forms of food, such as ice cream/sherbet, pudding/jelly, and biscuits/cookies, were also included in the questionnaire to assess the relationship between food preference and the hardness of each food, especially in post-metabolic/bariatric surgery cases and in elderly individuals. (5) Preference for each food item was rated on a visual analog scale (0: not at all, 10: extremely), with a picture of the food displayed to allow for associations through visualization. All responses took approximately 5 min to complete.

### 2.3. Assessment of Food Preference

Using the JFPQ developed as described above, we assessed the food preferences of the non-obese and abdominal-obese groups. The total scores for each food component were calculated for the carbohydrate, fat, protein, and dietary fiber groups. Because the maximum score differed for each nutrient group, we calculated the percentage of the total score to the full score for each nutrient group.

### 2.4. Assessment of Eating Behaviors

Obesity-related eating behaviors were evaluated using the questionnaire from the Guideline for Obesity issued by the Japan Society for the Study of Obesity (JASSO) [6,7,8,9,23,24]. This method specifically identifies problems in the various eating behaviors of obese subjects. This questionnaire comprises 55 questions on seven major scales as follows: (1) Recognition for weight and constitution (e.g., “Do you think it is easier for you to gain weight than others?”); (2) External eating behavior (e.g., “If food smells and looks good, do you eat more than usual?”); (3) Emotional eating behavior (e.g., “Do you have the desire to eat when you are irritated?”); (4) Sense of hunger (e.g., “Do you become irritated when you feel hungry?”); (5) Eating style (e.g., “Do you eat fast?”); (6) Food preference/content (e.g., “Do you often eat meat?”); (7) Regularity of eating habits (e.g., “Is your dinner time too late at night?”). All items were rated on a scale from 1 (seldom) to 4 (very often). By scoring the responses to the questionnaire according to sex and constructing a diagram to confirm their characteristics, we can objectively grasp issues related to eating behavior and habits. Because the maximum score for some major scales, including “recognition of weight and constitution”, “external eating behavior”, “sense of hunger”, and “food preference/content”, differs between males and females, we calculated the percentage of the full score for each eating behavior.

### 2.5. Assessment of Dietary Intake

Dietary intake was assessed using a self-administered long-Food Frequency Questionnaire (FFQ) developed for Japanese individuals [25]. Food intake was calculated by multiplying the frequency of consumption (never, 1–3 times/month, 1–2 times/week, 3–4 times/week, 5–6 times/week, once/day, 1–2 times/day, 4–6 times/day, and 7 times/day or more) by the relative portion size (small, medium, and large). Participants were asked to report their dietary intake over the previous year.

### 2.6. Statistical Analysis

All values are presented as the mean ± standard deviation (SD) and the number of subjects (%). For comparative analysis between the non-obese and abdominal-obese groups, Student’s *t*-test was used for continuous variables, and Fisher’s exact test was used for categorical variables. Because there were significant differences in sex and age between the non-obese and abdominal-obese groups, analysis of covariance (ANCOVA) was performed to adjust for age and sex as covariates when comparing the eating behavior scores, food preference scores, and dietary intake derived from the FFQ between the two groups.

Pearson’s correlation coefficients were used to examine the correlation between food preference scores, as assessed by the JFPQ, and dietary intake, as assessed by the FFQ, for each nutrient group. Logistic regression analysis was also used to examine the relationship between food preference scores and the frequency or amount of food intake for each food. For this purpose, the frequency of intake (never, 1–3 times/month, 1–2 times/week, 3–4 times/week, 5–6 times/week, 1 time/day, 1–2 times/day, 4–5 times/day, and 7 times/day or more) on the FFQ was placed on an ordinal scale from 1 to 9, and the relative amounts (small, medium, and large) were placed on an ordinal scale from 1 to 3.

In all cases, two-tailed tests were used, and *p* values < 0.05 were considered statistically significant. All analyses were performed using JMP Statistical Discovery Software 17.0 (SAS Institute, Cary, NC, USA).

## 3. Results

### 3.1. Nutritional Characteristics and Classification of the Foods in the JFPQ

Figure 1 shows the JFPQ developed in this study, which consists of 30 food items consumed daily by Japanese people. The nutritional characteristics of the 25 foods in each nutrient group are shown in Table 1, and detailed information on the foods used to calculate the nutritional value of each item is presented in Appendix A. The carbohydrate-to-total energy ratio of the foods in the carbohydrate group ranged from 77.1% to 100.0%, with an average of 90.2% for the sweet carbohydrate group and 84.2% for the non-sweet carbohydrate group, meeting our criterion of over 70%. The fat-to-total energy ratio of foods in the fat group ranged from 47.2% to 57.6%, with an average of 52.9% for the sweet fat group and 50.8% for the non-sweet fat group, above 40%. The protein content of the foods in the protein group ranged from 31.9% to 58.6% of the protein-to-total energy ratio, which was greater than 30%. The foods in the dietary fiber group ranged from 9 to 56 kcal of energy and 1.4 to 4.1 g of dietary fiber per 100 g. For the sweet and non-sweet food subgroups, the sweet carbohydrate group ranged from 12 to 90 g of sugar per 100 g of edible portion, and the sweet fat group ranged from 20 to 55 g, both of which met our criteria of 10 g or more. In contrast, the non-sweet carbohydrate group ranged from 0 g to 4 g, and the non-sweet fat group ranged from 1 g to 7 g.

### 3.2. Clinical Characteristics of the Participants in the Non-Obese and Abdominal-Obese Groups

Thirty-eight non-obese and 30 abdominal-obese subjects were enrolled in this study. The clinical characteristics of the study participants are shown in Table 2. The mean age was 36.4 ± 8.7 years in the non-obese group and 45.9 ± 10.7 years in the abdominal-obese group, which was significantly greater in the abdominal-obese group. There were significantly more males in the abdominal-obese group. The mean BMI was 21.0 ± 2.0 kg/m^2^ in the non-obese group and 27.1 ± 2.9 kg/m^2^ in the abdominal-obese group, and the mean waist circumference was 72.8 ± 7.1 cm in the non-obese group and 96.0 ± 7.1 cm in the abdominal-obese group, both of which were significantly greater in the abdominal-obese group. Systolic blood pressure, diastolic blood pressure, and the number of subjects with hypertension were also significantly higher in the abdominal-obese group.

### 3.3. Comparison of Eating Behavior and Food Preference Scores Between the Non-Obese and Abdominal-Obese Groups

First, we compared the eating behavior of non-obese and abdominal-obese individuals via the JASSO Eating Behavior Questionnaire. As shown in Table 3, the scores of the abdominal-obese group were significantly higher than those of the non-obese group on all seven major scales, including “food preference/content” represented by questions such as “Do you like fatty foods?”, “Do you often eat meat?”, and “Do you often eat sweet breads?”.

Given the differences in eating behaviors between the abdominal-obese and non-obese groups, the JFPQ was subsequently used to examine detailed food preferences in each group. As shown in Table 4, food preferences for each nutrient group were assessed as a percentage of the total score, and food preferences for each food were assessed as a score. The results are also presented as a radar chart to objectively capture the strength and bias of food preferences (Appendix A). The scores for the carbohydrate, fat, and protein groups were significantly higher in the abdominal-obese group than in the non-obese group, and this result did not change after adjusting for age and sex. Although the scores for the dietary fiber group were also significantly higher in the abdominal-obese group than in the non-obese group according to the univariate analysis, the significance disappeared after adjustment for age and sex. Comparisons were also made for the scores for each food comprising each nutrient group (Table 4). Even after adjusting for age and sex, preferences for several foods were found to be significantly greater in the abdominal-obese group, including Japanese sweets (*p* = 0.027), soft drinks (*p* = 0.029) (the sweet carbohydrate group), udon/soba noodles (*p* = 0.030) (the non-sweet carbohydrate group), sweet chocolates (*p* = 0. 007) (sweet fat group), snacks (*p* < 0.001), high-fat ramen noodles (*p* = 0.002), hamburgers (*p* = 0.037), deep-fried foods (*p* = 0.003) (non-sweet fat group), fish (*p* = 0.009), meat (*p* = 0.008), and eggs (*p* < 0.001) (protein group). In contrast, in the dietary fiber group, no food intake was significantly different between the non-obese and abdominal-obese groups.

### 3.4. Comparison of Dietary Intake of Total Energy and Macronutrients Between Non-Obese and Abdominal-Obese Groups

The FFQ was then used to examine the dietary intake of total energy and macronutrients in the non-obese and abdominal-obese groups. As shown in Table 5, the daily intakes of total energy (*p* < 0.001), non-fiber carbohydrates (*p* < 0.001), fat (*p* < 0.001), protein (*p* < 0.001), and dietary fiber (*p* < 0.001) were significantly greater in the abdominal-obese group than in the non-obese group, even after adjusting for age and sex.

### 3.5. Correlations Between Food Preference Scores and Dietary Intake

Finally, we investigated whether food preferences are associated with actual food intake. To this end, we examined the relationship between food preference scores, assessed by the JFPQ, and dietary intake, assessed by the FFQ. Across all study participants, there were significant positive correlations between food preference scores and intake for all four nutrient groups: carbohydrates (*p* = 0.002), fat (*p* < 0.001), protein (*p* < 0.001), and dietary fiber (*p* < 0.001) (Figure 2A,D,G,J). We then analyzed the relationship between food preferences and intake separately in the non-obese and abdominal-obese groups. In the non-obese group, a significant positive correlation was observed only for dietary fiber (*p* = 0.013) (Figure 2B,E,H,K). In the abdominal-obese group, a significant positive correlation was observed for fat (*p* = 0.024), and a trend toward a positive correlation was noted for carbohydrates (*p* = 0.091), but not for protein or dietary fiber (Figure 2C,F,I,L).

Next, a detailed examination of individual foods was conducted for the carbohydrate and fat groups, and positive correlations were observed between food preference and intake in the abdominal-obese group. As shown in Table 6, six foods in the carbohydrate group and five in the fat group were common to both the JFPQ and FFQ, allowing us to analyze the relationship between food preference and the frequency or amount of actual intake of these foods. For the six foods in the carbohydrate group, no correlation was found between food preference scores and the frequency or amount of intake in the non-obese group. In contrast, in the abdominal-obese group, there were significant positive correlations between the food preference score and the intake of all six foods in the carbohydrate group, regardless of whether they were sweet: Japanese sweets (*p* < 0.001 for frequency and *p* = 0.024 for amount), fruits (*p* = 0.043 for amount), soft drinks (*p* = 0.008 for frequency), rice crackers (*p* < 0.001 for frequency and *p* = 0.001 for amount), white rice (*p* < 0.001 for amount), and udon/soba noodles (*p* = 0.025 for frequency). In the fat group, significant positive correlations were found for three foods in the non-obese group: sweet chocolate (*p* < 0.001 for frequency and *p* < 0.001 for amount), snacks (*p* < 0.001 for frequency and *p* < 0.001 for amount), and high-fat ramen noodles (*p* = 0.046 for frequency). Moreover, in the abdominal-obese group, positive correlations were observed for all five foods in the fat group: cakes (*p* = 0.007 for frequency), sweet chocolates (*p* = 0.002 for frequency), snacks (*p* < 0.001 for frequency and *p* < 0.001 for amount), high-fat ramen noodles (*p* = 0.015 for frequency and *p* < 0.001 for amount), and deep-fried foods (*p* = 0.046 for amount).

## 4. Discussion

We developed a nutrition-based food preference questionnaire consisting of food items ordinally eaten by Japanese people, which can be used in clinical settings, including nutritional counseling. Using this questionnaire, named the JFPQ, the present study revealed that food preferences for carbohydrates, fat, and protein, but not for dietary fiber, were significantly greater in the abdominal-obese subjects than in the non-obese subjects. Furthermore, in the abdominal-obese subjects, preferences for carbohydrates and fat were positively correlated with actual dietary intake.

The Leeds Food Preference Questionnaire (LFPQ) [26] has been widely used in various studies worldwide [27,28] as a tool for assessing food preferences. The LFPQ allows the assessment of food reward along two nutritional dimensions: fat (high or low) and taste (sweet or savory/non-sweet). Although a Japanese version of the LFPQ (LFPQ-J) was recently developed [29], the nutritional criteria for each category remain relatively unclear, and it cannot be used to comprehensively assess food preferences based on individual nutrients. Therefore, to properly assess the food preferences of Japanese people, we developed the JFPQ, which includes foods that Japanese people consume daily. Based on a detailed nutritional assessment, the JFPQ categorizes each food into four groups: carbohydrate, fat, protein, and dietary fiber, allowing the carbohydrate and fat groups to be further evaluated as sweet and non-sweet subgroups. The JFPQ is expected to be used in clinical settings, such as nutritional counseling for obese patients, as it is quick and easy to complete, and the results can be converted into radar charts to visualize respondents’ food preferences for each nutrient (Appendix A).

Previous studies on food preferences in obese individuals have primarily focused on preferences for high-fat and high-sugar foods. Several reports have described a greater preference for fat among obese individuals [10,11,12,13,14,15,16]. However, findings regarding sweet food preferences in obese individuals have been inconsistent. Some studies have reported that obese individuals have a greater preference for sweet foods than non-obese individuals [10,12,17], whereas others have reported no association [30,31] or even a lower preference for sweet foods in obese individuals [11]. In the present study, we used the newly developed JFPQ to comprehensively assess the differences in food preferences between non-obese and abdominal-obese subjects across the four nutritional groups in more detail. The results showed that the abdominal-obese subjects in this study had greater preferences for carbohydrates, fat, and protein than non-obese individuals. However, after adjusting for age and sex, no significant differences were observed in the preferences for dietary fiber. In addition, the abdominal-obese subjects showed greater preferences for both sweet and non-sweet carbohydrate- and fat-rich foods. These findings suggest that obese Japanese individuals have a greater preference for all major energy-producing nutrients, regardless of whether they are sweet or non-sweet.

Using the Eating Behavior Questionnaire, we previously reported significant differences in several eating behaviors, including “food preference/content”, “sense of hunger”, “external eating behavior”, and “regularity of eating habits”, between type 2 diabetic patients with and without visceral fat accumulation [8]. In the present study, we also confirmed that, compared with non-obese subjects, abdominal-obese subjects had significantly higher scores in the “food preference/content” section of the Eating Behavior Questionnaire. This section examines how often they literally consume various types of energy-dense foods, such as noodles, fast food, pastries, fatty foods, meats, snacks, and sweet foods. Thus, these findings suggest that obese individuals have a greater preference for major energy-producing nutrients, particularly carbohydrates and fat, which is reflected in their actual eating behaviors related to “food preference/content.”

We then examined the relationship between food preference scores assessed by the JFPQ and actual food intake assessed by the FFQ. In non-obese subjects, a positive correlation was found for the dietary fiber group, whereas in abdominal-obese subjects, positive correlations were observed for the fat and carbohydrate groups. The non-obese group, which primarily consisted of healthcare professionals with nutritional knowledge, possibly consumed foods from the dietary fiber group according to their knowledge-based preferences, as these low-energy foods do not pose health concerns. In contrast, abdominal-obese individuals appeared to consume foods from the carbohydrate and fat groups, regardless of whether they were sweet or non-sweet, based on their cravings. This may have contributed to the overeating of high-carbohydrate and/or high-fat foods, leading to subsequent weight gain or visceral fat accumulation. Previous studies have reported a positive association between preference and intake of high-fat foods in obese individuals [11,32]. Our study supports this finding and further suggests that a high preference for carbohydrates is also likely to lead to increased carbohydrate consumption in obese Japanese individuals.

With respect to individual foods common to the JFPQ and FFQ, positive correlations between food preference and intake in the fat group were observed for sweet chocolates and snacks, even in non-obese subjects. Nevertheless, positive correlations were observed for more items, such as cakes, high-fat ramen noodles, and deep-fried foods, in abdominal-obese subjects. More interestingly, in the carbohydrate group, no correlation was observed between preference and intake for either the sweet or non-sweet subgroups in non-obese individuals. In contrast, abdominal-obese individuals showed positive correlations for all foods in this group, including the sweet subgroup (Japanese sweets, fruits, and soft drinks) and the non-sweet subgroup (white rice, udon/soba noodles, and rice crackers). The positive correlation between food preference and intake of non-sweet carbohydrates in obese Japanese individuals is noteworthy because, as a staple food, white rice, which is a non-sweet carbohydrate source, accounts for a large proportion of the energy intake in the Japanese population [33]. Furthermore, several previous studies have shown that increased consumption of white rice is associated with an increased risk of incident diabetes in Asian populations [34,35,36]. Taken together, the results of this study suggest that a strong preference for carbohydrate-rich staple foods, such as white rice, in addition to several high-fat foods, directly influences eating behavior in obese Japanese individuals, which might contribute to the pathogenesis of obesity and the development of obesity-related diseases, including diabetes.

When interventions to prevent overeating and obesity are considered, it is important to objectively understand an individual’s food preferences. Previous studies, including ours, have reported that obese patients who were treated with glucagon-like peptide-1 (GLP-1) receptor agonists [7,23] or those who underwent metabolic/bariatric surgery [24] showed improvements in eating behaviors, as measured by the Eating Behavior Questionnaire, which was associated with weight loss. These findings indicate that this questionnaire is a useful tool for monitoring the effectiveness of weight loss interventions. Similarly, we believe that the JFPQ can be used to evaluate the effects of interventions on obesity and metabolic disorders by assessing changes in food preferences. For example, a low-energy diet intervention has been reported to reduce preferences for high-carbohydrate and high-fat diets in overweight and obese individuals [37]. Additionally, weight loss programs for Japanese obese individuals significantly improved fat taste thresholds [38], suggesting that weight loss interventions may alter food preferences. Consequently, changes in food preferences may serve as indicators of the effectiveness of therapeutic interventions for obesity. Monitoring food preferences over time with the JFPQ may then help identify effective strategies for each individual case. Future longitudinal studies with larger sample sizes are necessary to examine changes in food preferences following various weight loss interventions, such as dietary restrictions, anti-obesity medications, and metabolic/bariatric surgery.

There were several limitations in this study. First, each questionnaire was based on self-assessment, and the possibility of underestimation or overestimation cannot be ruled out. Second, the non-obese group comprised healthcare professionals, potentially limiting the generalizability of the results to the broader population. Third, because previous studies reported sex differences in food preferences [13,14,15], the lower number of women in the abdominal-obese group may have influenced the results. Fourth, the reproducibility and repeatability of the JFPQ were not fully evaluated, although these steps are essential to further establish the utility of the JFPQ. Further research on food preferences, as assessed by the JFPQ in a larger general population, is needed to validate the present findings.

## 5. Conclusions

In summary, Japanese abdominal-obese individuals presented greater food preferences, particularly for fat- and carbohydrate-rich foods, which might be associated with increases in actual intake. This finding from our pilot study highlights the importance of considering food preferences in the pathogenesis of obesity. The JFPQ can help identify foods that trigger overeating based on an individual’s food preferences and may have potential clinical applications in a tailored approach to weight loss interventions.

## Figures and Tables

**Figure 1 nutrients-16-04252-f001:**
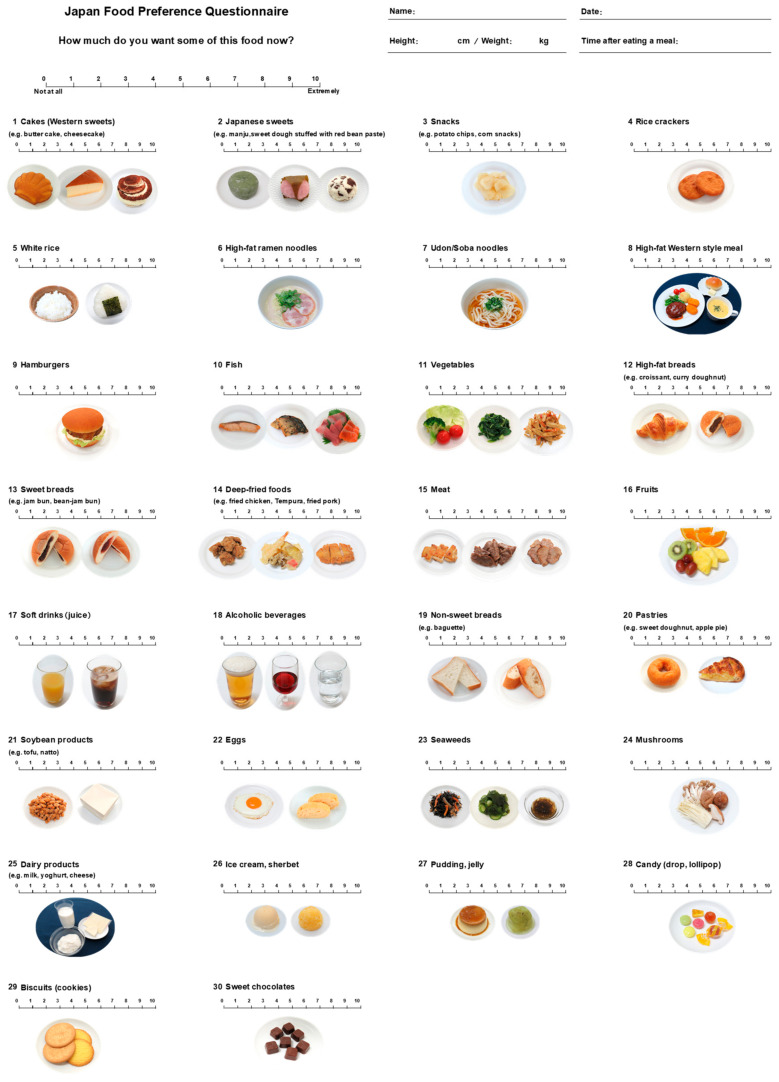
The Japan Food Preference Questionnaire (JFPQ). A newly developed Japan Food Preference Questionnaire (JFPQ) is presented. The JFPQ consists of questions that require participants to quantify how much they would like to eat the food items using a visual analog scale (VAS) ranging from 0 to 10 (0: not at all, 10: extremely). In the questionnaire, images of the foods are displayed to help the participants recognize the items.

**Figure 2 nutrients-16-04252-f002:**
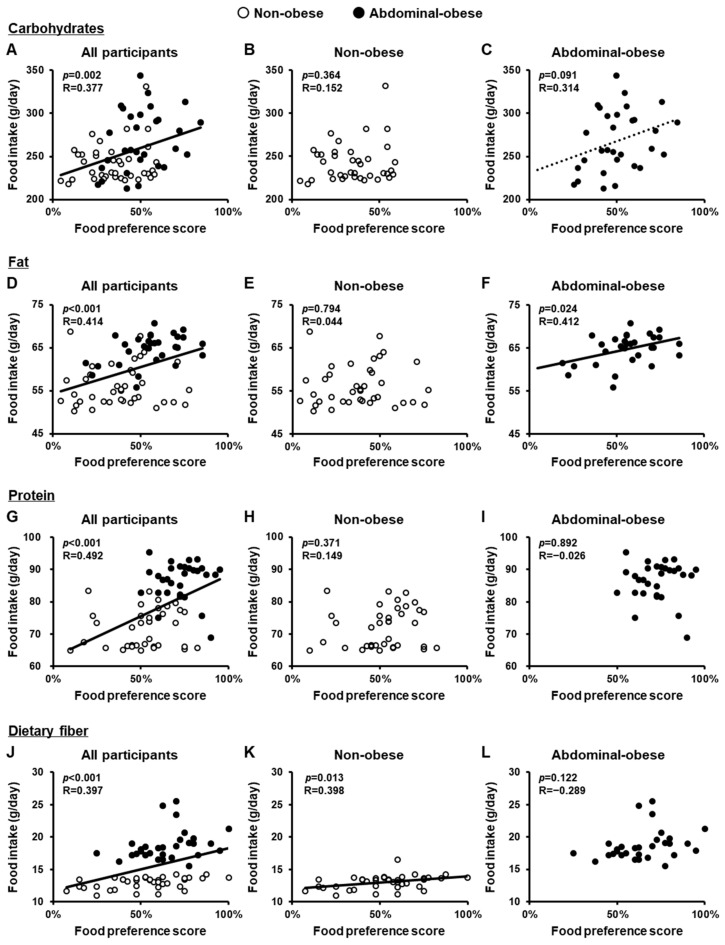
Correlations between food preference scores for each nutrient group, as assessed by the JFPQ, and daily dietary intake, as assessed by the FFQ. Food preference scores were assessed using the Japan Food Preference Questionnaire (JFPQ), and daily food intake (g/day) was assessed using the Food Frequency Questionnaire (FFQ) for each nutrient group. Correlations between food preference scores and dietary intake of carbohydrates (**A**–**C**), fat (**D**–**F**), protein (**G**–**I**), and dietary fiber (**J**–**L**) in all participants (**A**,**D**,**G**,**J**) and in the non-obese (**B**,**E**,**H**,**K**) and abdominal-obese (**C**,**F**,**I**,**L**) groups. *p* values were calculated using Pearson’s correlation coefficient.

**Table 1 nutrients-16-04252-t001:** Classification and nutritional characteristics of foods in each nutritional group of the JFPQ.

Carbohydrate Group	Criteria	Food	kcal /100 g	%Carb	%Fat	%Pro	Sugars g/100 g
Sweet	%Carb ≥ 70%	2. Japanese sweets	226	90.7	1.0	6.3	28
	Sugars ≥ 10 g/100 g edible portion	13. Sweet breads	269	78.4	11.9	8.3	24
		16. Fruits	56	84.1	1.8	3.9	12
		17. Soft drinks (juice)	46	97.6	1.0	1.3	10
		28. Candy (drop, lollipop)	385	100.0	0.0	0.0	90
		Mean	196	90.2	3.1	4.0	33
Non-sweet	%Carb ≥ 70%	4. Rice crackers	368	90.1	2.2	6.8	0
	Sugars < 10 g/100 g edible portion	5. White rice	156	91.6	1.2	5.1	0
		7. Udon/soba noodles	60	77.1	3.8	11.1	1
		19. Non-sweet breads	269	77.9	8.4	11.9	4
		Mean	213	84.2	3.9	8.8	1
Fat group	Criteria	Food	kcal /100 g	%Carb	%Fat	%Pro	Sugars g/100 g
Sweet	%Fat ≥ 40%	1. Cakes (Western sweets)	356	34.7	57.6	7.2	20
	Sugars ≥ 10 g/100 g edible portion	20. Pastries	337	45.7	47.5	5.9	18
		30. Sweet chocolates	541	38.3	56.2	3.7	49
		Mean	411	39.6	53.8	5.6	29
Non-sweet	%Fat ≥ 40%	3. Snacks	524	44.2	50.5	3.9	3
	Sugars < 10 g/100 g edible portion	6. High-fat ramen noodles	209	42.4	45.1	11.1	2
		8. High-fat Western-style meal	129	35.4	49.5	11.4	3
		9. Hamburgers	251	30.9	55.8	12.2	4
		12. High-fat breads	354	45.1	47.2	6.7	7
		14. Deep-fried foods	313	21.2	56.8	20.9	1
		Mean	297	36.5	50.8	11.0	3
Protein group	Criteria	Food	kcal /100 g	%Carb	%Fat	%Pro	
	%Pro ≥ 30%	10. Fish	127	14.8	26.5	58.6	
		15. Meat	186	6.7	57.3	36.0	
		21. Soybean products	106	9.0	51.0	35.0	
		22. Eggs	133	9.5	58.7	31.9	
		Mean	138	10.0	48.4	40.4	
Dietary fiber group	Criteria	Food	kcal /100 g	Dietary fiber g/100 g			
	Dietary fiber ≥ 1 g/100 g edible portion	11. Vegetables	32	3.2			
	Energy < 100 kcal/100 g edible portion	16. Fruits	56	1.4			
		23. Seaweeds	9	2.8			
		24. Mushrooms	27	4.1			
		Mean	31	2.9			

Carbohydrates refer to non-fiber carbohydrates that do not contain dietary fiber. %Carb: percentage of total energy from non-fiber carbohydrates; %Fat: percentage of total energy from fat; %Pro: percentage of total energy from protein. Sugars include glucose, fructose, galactose, sucrose, maltose, lactose, and trehalose. The data are presented as means. Detailed information on the foods used to calculate the nutritional value of each item is provided in Appendix A. JFPQ, Japan Food Preference Questionnaire.

**Table 2 nutrients-16-04252-t002:** Clinical characteristics of the participants in the non-obese and abdominal-obese groups.

	Non-Obese (N = 38)	Abdominal-Obese (N = 30)	*p* Value
Age, year	36.4 ± 8.7	45.9 ± 10.7	<0.001
Sex (male/female)	19/19	24/6	0.013
Body weight, kg	56.9 ± 8.9	79.7 ± 11.3	<0.001
BMI, kg/cm^2^	21.0 ± 2.0	27.1 ± 2.9	<0.001
Waist circumference, cm	72.8 ± 7.1	96.0 ± 7.1	<0.001
Systolic blood pressure, mmHg	116.6 ± 11.9	134.3 ± 13.4	<0.001
Diastolic blood pressure, mmHg	70.8 ± 8.5	87.1 ± 8.9	<0.001
Hypertension, *n* (%)	1 (2.6)	10 (33.3)	<0.001
Hyperlipidemia, *n* (%)	4 (10.5)	7 (23.3)	0.194
Diabetes, *n* (%)	0 (0.0)	3 (10.0)	0.081

The data are presented as the mean ± SD or number of subjects (percentage of total). *p* values were calculated using Student’s *t*-test for continuous variables and Fisher’s exact test for categorical variables. BMI, body mass index.

**Table 3 nutrients-16-04252-t003:** Comparison of eating behaviors between the non-obese and abdominal-obese groups.

	Non-Obese (N = 38)	Abdominal-Obese (N = 30)	Unadjusted	Sex and Age-Adjusted
			*p* Value	*p* Value
Recognition for weight and constitution (%)	45.0 ± 11.0	56.3 ± 14.1	<0.001	<0.001
External eating behavior (%)	42.8 ± 13.5	59.5 ± 15.1	<0.001	<0.001
Emotional eating behavior (%)	37.2 ± 13.5	47.5 ± 17.6	0.008	0.005
Sense of hunger (%)	41.0 ± 14.6	55.1 ± 12.3	<0.001	<0.001
Eating style (%)	43.3 ± 16.6	55.2 ± 15.1	0.003	<0.001
Food preference/content (%)	44.3 ± 11.7	56.1 ± 11.8	<0.001	<0.001
Regularity of eating habits (%)	48.7 ± 11.7	60.9 ± 13.6	<0.001	<0.001
Total score (%)	43.7 ± 8.9	56.7 ± 10.7	<0.001	<0.001

The data are presented as the mean ± SD. *p* values were calculated using Student’s *t*-test. Multiple regression analysis was performed using age and sex as covariates.

**Table 4 nutrients-16-04252-t004:** Comparison of food preference scores as assessed by the JFPQ between the non-obese and abdominal-obese groups.

	Non-Obese (N = 38)	Abdominal-Obese (N = 30)	Unadjusted	Sex and Age-Adjusted
			*p* Value	*p* Value
Carbohydrate group (%)	34.8 ± 15.5	50.5 ± 15.2	<0.001	0.007
Sweet (%)	28.2 ± 16.5	43.8 ± 17.6	<0.001	0.011
2. Japanese sweets	2.9 ± 2.8	5.3 ± 3.5	0.002	0.027
13. Sweet breads	2.4 ± 2.6	4.0 ± 2.5	0.014	0.174
16. Fruits	5.6 ± 3.0	6.5 ± 2.6	0.162	0.358
17. Soft drinks (juice)	2.1 ± 2.8	4.3 ± 3.0	0.003	0.029
28. Candy (drop, lollipop)	1.2 ± 1.6	1.8 ± 2.1	0.130	0.215
Non-sweet (%)	43.0 ± 18.5	58.9 ± 16.0	<0.001	0.023
4. Rice cracker	2.8 ± 2.6	4.6 ± 2.6	0.006	0.053
5. White rice	5.8 ± 2.8	7.5 ± 2.1	0.009	0.106
7. Udon/Soba noodles	5.0 ± 2.5	7.0 ± 2.5	0.002	0.030
19. Non-sweet breads	3.6 ± 3.1	4.5 ± 3.0	0.219	0.705
Fat group (%)	37.9 ± 19.5	55.5 ± 16.9	<0.001	0.002
Sweet (%)	39.9 ± 29.9	56.0 ± 24.1	0.019	0.042
1. Cakes (Western sweets)	4.4 ± 3.4	6.2 ± 3.4	0.039	0.074
20. Pastries	3.5 ± 3.2	4.6 ± 2.5	0.151	0.414
30. Sweet chocolates	4.0 ± 3.3	6.0 ± 2.8	0.008	0.007
Non-sweet (%)	36.9 ± 18.5	55.3 ± 19.6	<0.001	0.001
3. Snacks	3.0 ± 2.9	5.7 ± 2.9	<0.001	<0.001
6. High-fat ramen noodles	4.1 ± 2.8	6.8 ± 2.8	<0.001	0.002
8. High-fat Western-style meal	4.4 ± 2.4	5.2 ± 2.7	0.254	0.662
9. Hamburgers	2.8 ± 2.8	4.5 ± 3.3	0.023	0.037
12. High-fat breads	3.4 ± 2.7	4.4 ± 2.9	0.125	0.278
14. Deep-fried foods	4.4 ± 2.5	6.6 ± 2.3	<0.001	0.003
Protein group (%)	51.8 ± 17.1	72.8 ± 11.7	<0.001	<0.001
10. Fish	5.3 ± 2.2	7.2 ± 2.3	0.001	0.009
15. Meat	5.6 ± 2.5	7.4 ± 2.5	0.003	0.008
21. Soybean products	4.9 ± 3.0	7.1 ± 2.2	0.002	0.053
22. Eggs	4.9 ± 2.3	7.4 ± 1.9	<0.001	<0.001
Dietary fiber group (%)	52.6 ± 21.5	64.6 ± 17.1	0.016	0.200
11. Vegetables	6.4 ± 2.3	7.3 ± 2.2	0.106	0.348
16. Fruits	5.6 ± 3.0	6.5 ± 2.6	0.162	0.358
23. Seaweeds	4.5 ± 2.5	6.3 ± 2.4	0.004	0.089
24. Mushrooms	4.6 ± 2.9	5.7 ± 2.7	0.112	0.729

The data are presented as the means ± SDs. Food preferences for each nutrient group are presented as percentages of the total score to the full score, and the preferences for each food item are presented as scores. *p* values were calculated using Student’s *t*-test. Multiple regression analysis was performed with age and sex as covariates. JFPQ, Japan Food Preference Questionnaire.

**Table 5 nutrients-16-04252-t005:** Comparison of daily energy and macronutrient intake as assessed by the FFQ between the non-obese and abdominal-obese groups.

	Non-Obese (N = 38)	Abdominal-Obese (N = 30)	Unadjusted	Sex and Age-Adjusted
			*p* Value	*p* Value
Energy, kcal/day	1824.5 ± 187.4	2187.3 ± 206.6	<0.001	<0.001
Non-fiber carbohydrate, g/day	242.6 ± 22.9	268.3 ± 35.3	<0.001	0.034
Fat, g/day	56.2 ± 4.8	64.7 ± 3.5	<0.001	<0.001
Protein, g/day	71.9 ± 6.1	86.5 ± 5.9	<0.001	<0.001
Dietary fiber, g/day	13.0 ± 1.1	18.7 ± 2.4	<0.001	<0.001

Daily energy and macronutrient intakes were assessed using the Food Frequency Questionnaire (FFQ). The data are presented as the mean ± SD. *p* values were calculated using Student’s *t*-test. Multiple regression analysis was performed with age and sex as covariates.

**Table 6 nutrients-16-04252-t006:** Correlations between food preference scores assessed by the JFPQ and frequency or amount of intake assessed by the FFQ.

Food Preference Score by JFPQ	Frequency or Amount of Intake by FFQ	Non-Obese (N = 38)	Abdominal-Obese (N = 30)
		*p* Value	*p* Value
Carbohydrate/Sweet			
2. Japanese sweets	Frequency	0.539	<0.001
	Amount	0.644	0.024
16. Fruits	Amount	0.093	0.043
17. Soft drinks	Frequency	0.253	0.008
Carbohydrate/Non-Sweet			
4. Rice crackers	Frequency	0.115	<0.001
	Amount	0.237	0.001
5. White rice	Frequency	0.125	0.058
	Amount	0.063	<0.001
7. Udon/soba noodles	Frequency	0.183	0.025
	Amount	0.771	0.067
Fat/sweet			
1. Cakes (Western sweets)	Frequency	0.497	0.007
	Amount	0.242	0.081
30. Sweet chocolates	Frequency	<0.001	0.002
	Amount	<0.001	0.238
Fat/non-sweet			
3. Snacks	Frequency	<0.001	<0.001
	Amount	<0.001	<0.001
6. High-fat ramen noodles	Frequency	0.046	0.015
	Amount	0.069	<0.001
14. Deep-fried foods	Frequency (Deep-fried pork)	0.737	0.638
	Amount (Deep-fried pork)	0.397	0.046

Food preference scores were assessed using the Japan Food Preference Questionnaire (JFPQ). The frequency and amount of intake were assessed using the Food Frequency Questionnaire (FFQ). *p* values were calculated using logistic regression analysis.

## Data Availability

The data presented in this study are available from the corresponding author upon reasonable request due to the privacy of research participants.

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
