# Peer review of "Food Preference Assessed by the Newly Developed Nutrition-Based Japan Food Preference Questionnaire and Its Association with Dietary Intake in Abdominal-Obese Subjects"

_nutrients, 2024, doi:10.3390/nu16234252_

Round 1
Reviewer 1 Report
Comments and Suggestions for Authors
This work developed a consisting of food items and evaluated the differences in food preferences between non-obese and abdominal-obese individuals by using this questionnaire. Additionally, the relationships between these preferences and actual dietary intake were investigated. The experiments were well performed, and the results were presented logically.
1. Title: the ‘newly developed Japan Food Preference Questionnaire’ is not clear. It may be better to change it with some characteristics, such as ‘nutrition-based food preference questionnaire’.
2. How to adjust the eating behavior scores, food preference scores, and dietary intake calculated from the FFQ for age and sex?
Author Response
Response to Reviewer #1
Reviewer comments for the author
This work developed a consisting of food items and evaluated the differences in food preferences between non-obese and abdominal-obese individuals by using this questionnaire. Additionally, the relationships between these preferences and actual dietary intake were investigated. The experiments were well performed, and the results were presented logically.
Authors’ response
We thank you for your time and effort in reviewing our manuscript. We really appreciate beneficial suggestions to improve the quality of our manuscript. We have responded to your comments point by point as follows:
Comments 1: Title: the ‘newly developed Japan Food Preference Questionnaire’ is not clear. It may be better to change it with some characteristics, such as ‘nutrition-based food preference questionnaire’.
Authors’ response
Thank you for your suggestion. We agree that the title could be more descriptive to better reflect the nutritional focus of the questionnaire. We propose modifying the title to: “Food preference assessed by the newly developed nutrition-based Japan Food Preference Questionnaire and its association with dietary intake in abdominal-obese subjects.” We have also revised the Abstract section as follows:
This revision should highlight the nutritional aspects of the questionnaire.
- On page 1, in the Abstract section
“We developed the Japan Food Preference Questionnaire (JFPQ) to evaluate food preferences across four nutrient groups based on nutritional evidence: carbohydrate, fat, protein, and dietary fiber. ”
Comments 2: How to adjust the eating behavior scores, food preference scores, and dietary intake calculated from the FFQ for age and sex?
Authors’ response
Thank you for your important comment. We used statistical adjustments for age and sex when analyzing the differences in the food preference scores, eating behavior scores, and dietary intake calculated from the FFQ between the non-obese and abdominal-obese participants. Specifically, analysis of covariance (ANCOVA) was employed to account for these variables. To address this concern, we have revised the manuscript as follows:
- On page 4, in the Materials and Methods section (2.6. Statistical analysis)
“Because there were significant differences in gender and age between the non-obese and abdominal-obese groups, analysis of covariance (ANCOVA) was performed to adjust for age and sex as covariates when comparing the eating behavior scores, food preference scores, and dietary intake derived from the FFQ between the two groups.”
We hope these modifications address your concerns and improve the overall quality of our work. Thank you once again for your valuable feedback.

Reviewer 2 Report
Comments and Suggestions for Authors
The authors stated the aim of this study was to develop a new food preference questionnaire based on detailed nutritional characteristics, consisting of food items ordinally eaten by Japanese people. Hence, the developed tool concerns a specific population group and specific diet.
My doubts concern the lack of verification of the repeatability of the questionnaire and the consistency of the data with the results collected in a direct interview or by the current recording method.
Therefore, I propose to add in the title and content that this is a pilot study.
Author Response
Response to Reviewer #2
Reviewer comments for the author
The authors stated the aim of this study was to develop a new food preference questionnaire based on detailed nutritional characteristics, consisting of food items ordinally eaten by Japanese people. Hence, the developed tool concerns a specific population group and specific diet.
My doubts concern the lack of verification of the repeatability of the questionnaire and the consistency of the data with the results collected in a direct interview or by the current recording method.
Therefore, I propose to add in the title and content that this is a pilot study.
Authors’ response
Thank you for your valuable comment. We completely agree with the reviewer’s important concerns. As this study represents an initial evaluation of a newly developed questionnaire, JFPQ, its primary focus was on application and its association with dietary intake. However, we acknowledge that the study did not include an adequate validation of reproducibility (test-retest reliability). This important limitation has been addressed in the “Discussion” section and noted as a priority for future research.
In response to the reviewer’s suggestion, we have also revised the Abstract and Conclusions to clearly indicate that this study is a “pilot study” and to emphasize its exploratory nature. The following revisions have been made to the manuscript.
- On page 1, in the Abstract section
“Our findings from this pilot study demonstrated that abdominal-obese individuals had greater preferences for fat and carbohydrates, which were linked to actual fat and carbohydrate intake and possibly contributed to the development of obesity.”
- On page 14, in the Discussion section
“Fourth, the reproducibility and repeatability of the JFPQ were not fully evaluated, although these steps are essential to further establish the utility of the JFPQ.”
- On page 14, in the Conclusions section
“This finding from our pilot study highlights the importance of considering food preferences in the pathogenesis of obesity.”
These revisions aim to clarify the exploratory scope of the study and address the reviewer's concerns. Thank you for highlighting these important issues, which will improve the quality of the manuscript.
